# *De Novo* Polycomb Recruitment and Repressive Domain Formation

Itzel Alejandra Hernández-Romero and Victor Julian Valdes *

Department of Cell Biology and Development, Institute of Cellular Physiology (IFC), National Autonomous University of Mexico (UNAM), Mexico City 04510, Mexico
* Correspondence: julian.valdes@ifc.unam.mx

**Abstract:** Every cell of an organism shares the same genome; even so, each cellular lineage owns a different transcriptome and proteome. The Polycomb group proteins (PcG) are essential regulators of gene repression patterning during development and homeostasis. However, it is unknown how the repressive complexes, PRC1 and PRC2, identify their targets and elicit new Polycomb domains during cell differentiation. Classical recruitment models consider the pre-existence of repressive histone marks; still, *de novo* target binding overcomes the absence of both H3K27me3 and H2AK119ub. The CpG islands (CGIs), non-core proteins, and RNA molecules are involved in Polycomb recruitment. Nonetheless, it is unclear how *de novo* targets are identified depending on the physiological context and developmental stage and which are the leading players stabilizing Polycomb complexes at domain nucleation sites. Here, we examine the features of *de novo* sites and the accessory elements bridging its recruitment and discuss the first steps of Polycomb domain formation and transcriptional regulation, comprehended by the experimental reconstruction of the repressive domains through time-resolved genomic analyses in mammals.

**Keywords:** polycomb targeting; *de novo* recruitment; binding; nucleation; spreading; repression; PRC1; PRC2

## 1. Introduction

The Polycomb domains are secondary structures in chromosome organization associated with chromatin compaction and gene repression [1–3]. The configuration of these chromatin domains varies within each cell lineage, and their main characteristic is the presence of the Polycomb protein complexes and two histone marks: the tri-methylation of histone H3 on lysine 27 (H3K27me3) and the monoubiquitylation of histone H2A on lysine 119 (H2AK119ub1) [4,5].

The Polycomb repressive complexes 1 and 2 (PRC1 and PRC2) are essential for cell viability and differentiation, as they maintain the repressed state of determinant genes during embryonic development and tissue homeostasis [6–10]. In mice, the loss of PRC1 leads to lethality at the two-cell stage [11]. In contrast, the ablation of PRC2 in mouse embryos causes severe defects during gastrulation, and organisms succumb around the seventh embryonic day when the cell executes fate decisions [12–15]. This behavior reflects the essential role of Polycomb in maintaining and rewiring the transcriptional repression network that sustains pluripotency and posterior lineage specification [16,17]. Similarly, alterations in Polycomb activity affect the culture of mouse embryonic stem cells (mESC) as the lack of PRC1 causes proliferation to cease and the loss of typical mESC morphology [18,19], whereas PRC2 deficiency allows self-renewal but no cell differentiation in vitro [16,20–22]. An estimated 4000 polycomb targeting sites in mESC dynamically switch upon cell-fate specification [23]. However, how specific the Polycomb targets are that are established during differentiation is a fundamental question in epigenetics.

PRC1 and PRC2 have an interlinked influence on their binding and catalytic activity, although they can be recruited independently to chromatin by sampling the chromatin

environment [24,25]. The passive sampling model proposes that the Polycomb proteins interact weakly and transiently with all their potential binding sites but only accumulate when the lack of antagonistic signals allows their residence [26,27]. Likewise, at least three axes reinforce Polycomb recruitment: (1) the catalysis and recognition of H3K27me3, as the hierarchical model poses [28,29], (2) the deposition and union of H2AK119ub, according to the alternative model [19,30], and (3) the *de novo* recruitment which is independent of repressive histone marks (Figure 1).

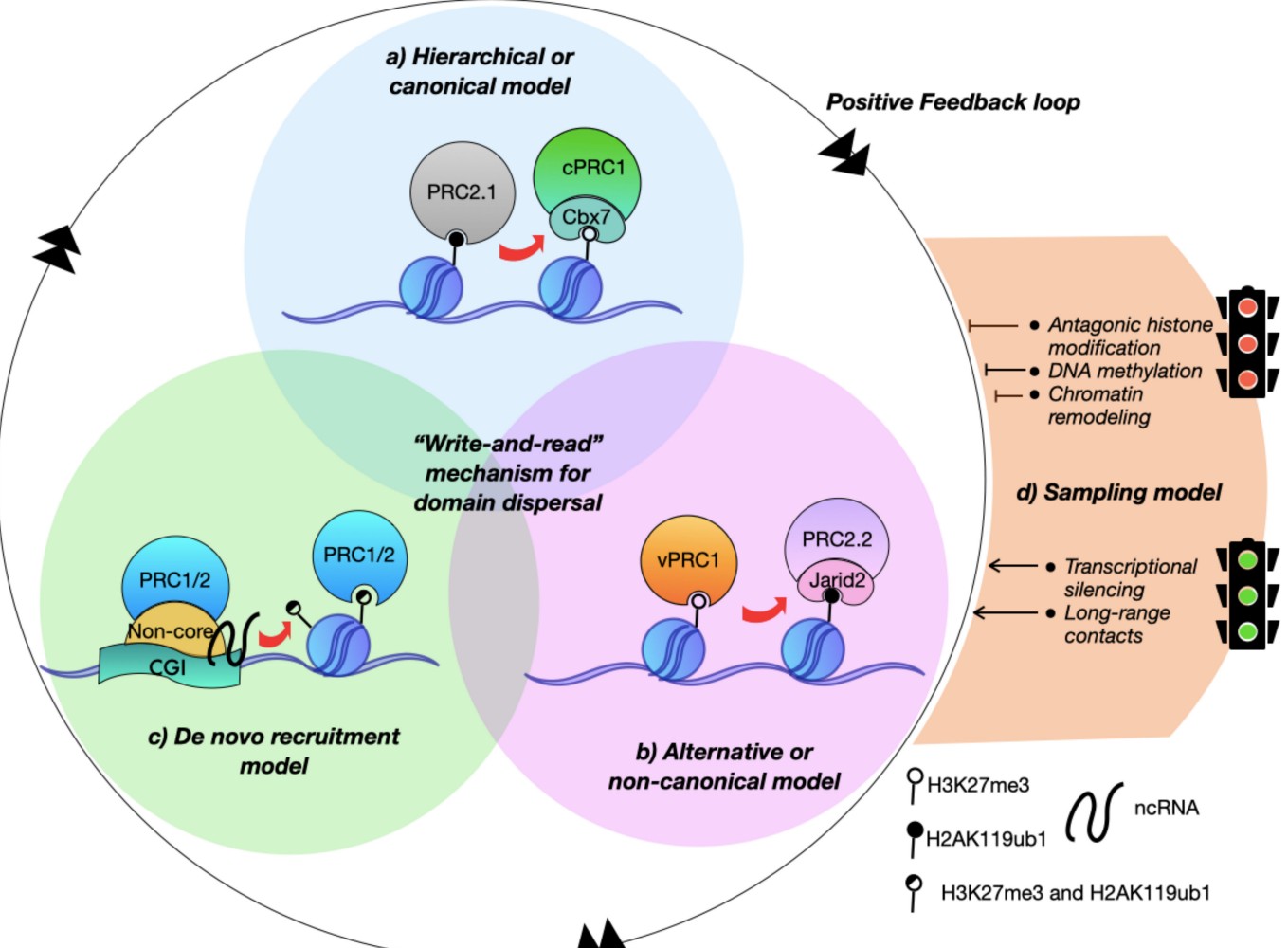

**Figure 1.** Polycomb recruitment models. Three recruitment models contribute to the Polycomb feedback loop that allows the inheritance and maintenance of the repressive domains. (**a**) The hierarchical or canonical recruitment model assumes that the subcomplex 1 of PRC2 (PRC2.1) binds to their targets, where the canonical PRC1 complex (cPRC1) joins H3K27me3-labelled chromatin; (**b**) the alternative or non-canonical recruitment model is centered on the subcomplex 2 of PRC2 (PRC2.2) recognition of H2AK119ub1 deposited by the variant PRC1 complex (vPRC1); (**c**) the *de novo* recruitment model reflects the capacity of PRC1 and PRC2 to bind its targets in the absence of the Polycomb repressive histone marks. In all cases, the write-and-read mechanism supports the replication inheritance, maintenance, and dispersal of the H3K27 methylation; (**d**) the binding of Polycomb complexes is influenced by the local chromatin environment and transcriptional status.

Once the Polycomb group proteins (PcG) are attached to the chromatin, the write-and-read mechanism of histone methylation perpetuates the inheritance. The positive feedback loop guides the dispersal and preservation of the Polycomb repressive domains to maintain cellular identity [31]. However, the *de novo* recruitment is the only signal

that overcomes the dysregulation of the cellular Polycomb steady-state condition and the principal recruitment mechanism initiating the reconfiguration of the repressive patterns during cell differentiation and development.

A remarkable characteristic of Polycomb targeting is that the *de novo* recruitment signal can be independent of the catalytic activity of the complexes; this was demonstrated in PRC1/2 reintroduction experiments that resulted in the accurate reconstitution of the lineage-specific H3K27me3 or H2AK119ub1 patterns [22,31,32]. Therefore, *de novo* recruitment sites are specific and constant after cell division but dynamic during differentiation. However, the mechanism by which Polycomb can find them is still hotly debated. After all, what is the *de novo* signal at the nucleation sites? How does *de novo* recruitment prevail in the chromatin despite the absence of the Polycomb-mediated histone modifications?

Thus, besides the exhaustive and successful efforts to characterize the Polycomb domains, most of the evidence related to recruitment has come from genomic analyses that reflect the steady-state conditions or the prolonged effects of Polycomb depletion. Unfortunately, these approaches conceal the initial steps in Polycomb binding and domain dispersal, such as the prior effects in gene expression. More recently, methods that assess the active turnover of the histone modifications and *de novo* recruitment have become particularly attractive, such as inducible genetic editing technologies, degron tag-based approaches, and the usage of inhibitors allowing the rapid suppression and reintroduction of the PcG subunits and therefore, the study of the establishment of repressive domains through their restoration in time. In this review, we discuss the most accepted Polycomb binding models in mammals and focus on the new findings of Polycomb *de novo* recruitment and repressive domain formation in light of the latest time scale analysis in the absence of repressive marks. We also examine the features of *de novo* sites, the role of accessory elements attained to these sites, and how Polycomb catalytic activity and three-dimensional arrangements inside the nucleus maintain the regulatory feedback loop that supports lineage specification in the context of recruitment.

## 2. Polycomb Nucleation Sites Concur with CpG Islands

The Polycomb members and their target genes are highly conserved, but their recruitment mechanisms can diverge considerably [33,34]. For example, in *Drosophila*, PRC2 binds to conserved DNA sequences known as Polycomb Response Elements (PRE) with specific sequences recognized by DNA-binding proteins such as Zeste, Gaga, and Pho [35]. Pho is essential for Polycomb recruitment and repressive domain nucleation [36]. Similarly, the role of some mammalian transcription factors in the recruitment of Polycomb arose in the field, such as Rest, Runx1, Yy1, and Snail [37–40]. Nonetheless, the locus-specific evidence does not hint at the existence of mammalian Polycomb consensus motif. Moreover, evidence has emerged that neither PRE nor the binding profiles of PRC2-associated transcription factors are sufficient to wholly predict the PRC2 localization in the *Drosophila* genome [41–44].

At mammalian promoters, the enrichment of interspaced CG dinucleotide sequences, known as CpG islands (CGIs), is the only widely reported and accepted feature of Polycomb binding [6,7,24,45,46]. Consequently, inserting GC-rich elements in free signal environments can recruit PRC2 in mESC [24,47–49]. However, CGI alone cannot per se explain all the diversity of Polycomb targets as there is a set of endogenous CGIs with no differences in length, CpG position, GC density, or content that are avoided by Polycomb, pointing out the existence of additional elements that regulate Polycomb *de novo* recruitment in a tissue and temporal-specific manner [17,50]. The specificity is partially associated with the DNA methylation antagonism because hypomethylation does not affect the catalytic activity of PRC1/2 but causes new binding to non-endogenous Polycomb sites and reduces binding to their usual targets, probably by complex dilution [47,48,51–53]. Recent data have shown that in mESC, the nucleation sites within the CGIs are over-represented by one tandem repeat motif rich in "GA" or "GCN" that lies near the transcription start site (TSS) of Polycomb-targeted genes [31]. The hypothesis is that the intrinsic affinity of Polycomb

complexes for these non-methylated DNA sequences, in addition to the epigenetic environment, allows its accessibility and regulates a lineage-specific *de novo* recruitment [54]. However, it is evident that CGI alone cannot exclusively explain PcG recruitment based on the diversity of Polycomb-decorated sites over different cell lineages and tissues. Therefore, other molecular players and mechanisms operate in *de novo* Polycomb recruitment.

## 3. Accessory Elements Guide *De Novo* Recruitment and Nucleation Site Formation

The Polycomb repressive complexes possess various core and non-core subunits that are dynamically regulated depending on the physiological or developmental state [55,56]. Although none of the core subunits directly recognizes a specific DNA motif, accessory molecules can bind CG-rich sequences [27,57]. In the next section, we will discuss the participation of Polycomb non-core proteins and RNA in the *de novo* attachment to Polycomb nucleation sites.

### 3.1. The PRC1 Variant Complexes Are Essential to Nucleate Polycomb De Novo Target Sites

The functional core of PRC1 in mammals has one of the six Polycomb group ring finger proteins (Pcgf1-6) dimerized with a Ring finger protein subunit (Ring1A or Ring1B), which possesses E3 ligase activity [58–60]. PRC1 is subclassified into canonical (cPRC1) and variant complexes (vPRC1) depending on the assembly of Pcgf and Chromobox (Cbx) subunits [61–63]. The two cPRC1 complexes, PRC1.2 and PRC1.4, recognize H3K27me3 through a Cbx subunit to mediate chromatin compaction and long-range interactions [19,64]. In contrast, the four vPRC1 complexes (PRC1.1, PRC1.3, PRC1.5, and PRC1.6) bind to chromatin in a Cbx-independent manner and they are directly associated with the enhanced deposition of H2AK119ub1 [65].

Most evidence indicates the direct participation of vPRC1, but not cPRC1, in Polycomb recruitment. The minor contribution of cPRC1 on PRC2 recruitment is explained by its lower E3 ubiquitin ligase activity compared to vPRC1 [19,61,66–68]. The conditional ablation of Pcgf2 in Pcgf4-deficient mESC showed no changes in gene expression, Suppressor of zeste subunit (Suz12) binding, or the deposition of H2AK119ub1 and H3K27me3, suggesting that cPRC1 is not essential for Polycomb *de novo* recruitment [67]. In line with these results, inducible ablation systems have shown that catalytical dead forms of Ring1B do not recruit PRC2 [19,69]. Thus, despite the potential capacity of cPRC1 to bind *de novo* spots through different non-core subunits, only those in the vPRC1 may be able to initiate nucleation site formation at endogenous targets through an augmented H2AK119ub1 deposition.

Inducible recruitment systems in artificial chromatin environments support the notion that only the complexes containing Pcgf1, Pcgf3, Pcgf5, and Pcgf6 can recruit Enhancer of zeste methyltransferase 2 (Ezh2) and Suz12, leading to a significant increase in H2AK119ub1 and H3K27me3 [19,70]. vPRC1 complexes containing Pcgf1, Pcgf3, and Pcgf6 are the least affected by the perturbation of the positive feedback loop due to their interaction with DNA-binding proteins (Figure 2). For example, the reintroduction of catalytic dead Ring1A/B proteins does not affect the genome-wide binding of Pcgf1 and Pcgf6, which associates with Kdm2b and Max factor, among others [70–73]. In particular, the demethylase Kdm2b binds CGIs via its Zinc finger motif [30] and interacts with Pcgf1, the vPRC1 subunit with the highest chromatin-bound fraction in mES [27] (Figure 2). Even the catalytically inactive version of Kdm2b fused to a Tet repressor (TetR) can recruit the PRC1 to Tet operator (TetO) if Pcgf1 is part of the complex [19]. Hence, the participation of Kdm2b in *de novo* recruitment is independent of its catalytic activity but dependent on Pcgf1. Consequently, the inducible removal of Pcgf1 causes considerable H2AK119ub1 and H3K27me3 reductions and the reactivation of hundreds of Polycomb targets in mESC [67]. In contrast, the inducible impairment of PRC2 recruitment in Pcgf1-null cells frustrates cPRC1 and vPRC1 targeting and prolonged culturing [72,74]. All this evidence points out that Pcgf1-mediated recruitment is critical for the Polycomb feedback loop.

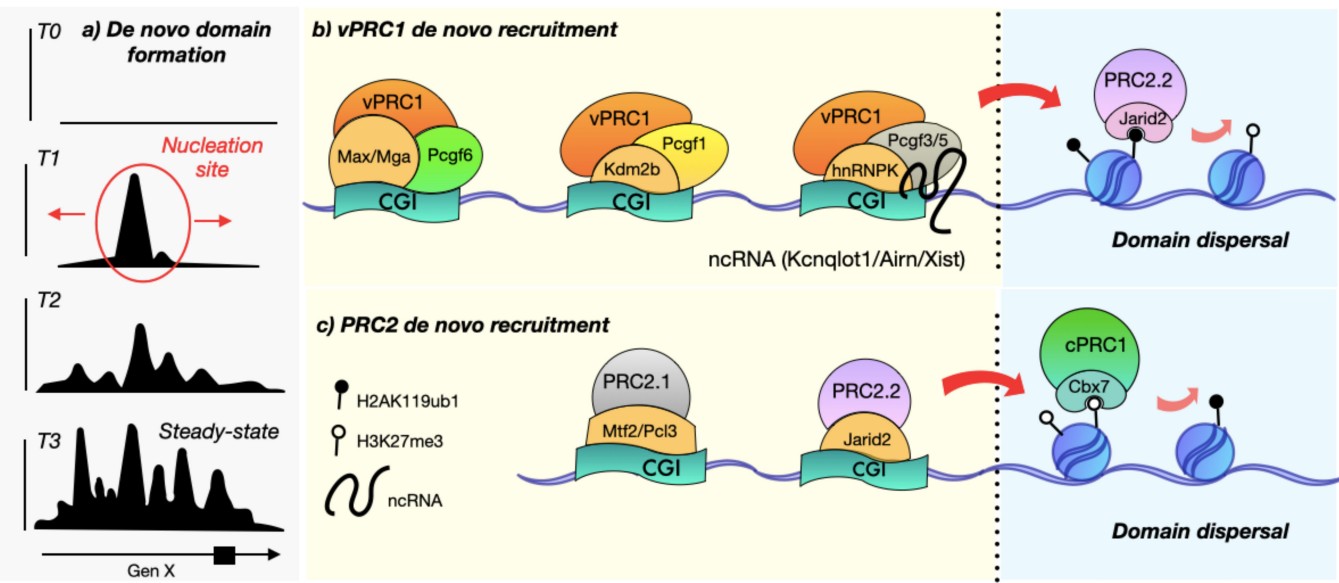

**Figure 2.** Polycomb *de novo* recruitment seeds the domain nucleation signal. The chromatin occupancy of Polycomb progress after its reintroduction in virtue of time. (**a**) *De novo* domain formation: different Polycomb complexes are recruited to specific sites during development and differentiation. Experimental reconstitution assays show that the first Polycomb recruitment sites coincide with the nucleation sites that spread the large repressive domains observed on the steady-state condition. The *y*-axis represents a hypothetical peak height and the *x*-axis represents the Polycomb target gene (gene X) and its promoter as a box. (**b**) The vPRC1 complexes interact with DNA-binding proteins through the Pcgf subunits and non-coding RNA (ncRNAs). (**c**) The PRC2 complexes bind their targets at CpG islands (CGIs) by interacting with Pcl and Jarid2 subunits. Once established, the nucleation sites allow the domain dispersal through the positive feedback loop. Please refer to Table 1 for more information about the role of each subunit.

There is also evidence of Pcgf1 participation in *de novo* recruitment during development, as an expression of the Kdm2b long isoform (Kdm2bl) at the mouse peri-implantation stage correlated with the *de novo* Pcgf1 recruitment. In this case, the recruitment occurs after the massive post-fertilization erasure of H3K27me3 that allows the repressive domain to be established at the post-implantation stages [30]. Thus, it is an example of how the time-specific expression of non-core subunits and their isoforms are involved in *de novo* recruitment.

Another vPRC1 complex, PRC1.3, is necessary for cell reprogramming and thus potentially relevant for *de novo* recruitment. Pcgf3 interacts with Nrf1, Fibrosin, Prdm14, Usf1/2, or Auts2 depending on the primed stage to naïve conversion of pluripotent cells [75–77]. Interestingly, the Prdm14-AID inducible degradation causes transcriptional derepression in primordial germ cells [75], the knockdown of Usf1/2 factors displaces PRC1.3 binding in mESC [76], and Nrf1 directs PRC1.3 to essential neurodevelopmental genes [77]. These support the hypothesis that non-Polycomb accessory proteins recruit Pcgf3 to their targets in cell reprogramming [75].

The last well-documented example of Pcgf participation in vPRC1 *de novo* recruitment is Pcgf6, which binds its targets despite the loss of H3K27me3 [70], presumably by the association with chromatin remodelers and DNA-binding proteins in mESC, such as G9a, L3mbtl2, Max, Mga, E2f6, and Dp1, explaining the closer proximity of PRC1.6 to TSS and enrichment at shorter CGIs compared to cPRC1 complexes [70,78–81]. In particular, the capability of the Max factor to favor domain nucleation has been verified in artificial recruitment sites, where its binding correlates with Pcgf6, H2AK119ub1, and H3K27me3 histone mark enrichment [70] (Figure 2). Pcgf6 is crucial for embryo implantation, cell viability, and the repression of germline and meiotic genes [70,82,83]. Thus, the above highlights the participation of DNA-binding proteins in Pcgf recruitment to vPRC1 *de novo*

sites. In all likelihood, more evidence regarding the role of accessory proteins in vPRC1 will arise in future tissue-specific studies.

### 3.2. PRC2 Creates Narrow De Novo Recruitment Sites

The functional core of PRC2 contains the embryonic ectoderm development protein (Eed), the retinoblastoma binding protein (RbAp46/48), Suz12, and Ezh1/2 that are responsible for the catalysis of mono-, di-, and trimethylation of histone H3 lysine 27 (H3K27me1/2/3) [4,84]. Within PRC2 complexes, Suz12 is the structural platform that coordinates the assembly of different core and non-core subunits [85–87]. The VEFS domain of Suz12 interacts with the PRC2 core subunits and is needed to sustain K27 methylation; meanwhile, the N terminal domain links the accessory subunits and is essential for PRC2 binding to the CGIs [4,88]. Interestingly, the binding pattern of Suz12 in mESC is unaffected by the loss of H3K27me3 caused by Ezh2-inhibitors or the Eed and Ezh1/2 double knockout [32]; moreover, the reintroduction of Ezh2 or Eed restores the original H3K27 methylation patterns as long as Suz12 is present [31,32], demonstrating that Suz12 is the fundamental platform for the ensemble and targeting of PRC2.

The PRC2 *de novo* recruitment can be independent of H2AK119ub1 availability under some circumstances, since core subunits binding at genome-wide levels persist despite reduced free ubiquitin levels in the cell [20,22,89]. Likewise, Suz12 can be recruited to the chromatin even when H3K27me3 and H2AK119ub1 are simultaneously unavailable, thus independently of the write-and-read mechanism. Indeed, the conditional reintroduction of Eed in mESC wherein ubiquitin pools were pharmacologically depleted showed that H2AK119ub1 reaccumulated at the same targets after Eed reintroduction and replenishment of ubiquitin. This evidence remarkably showed that the PRC2 recruitment signal was maintained in the absence of both repressive marks [22]. The latter was corroborated in mESC where: (1) KO of Ring1A, Ezh1, and Ezh2 occurred; (2) Ring1B was conditionally deficient for upon tamoxifen administration; and (3) they were subjected to the reintroduction of Ezh2 by a Doxy treatment [20]. Under these conditions, there were no changes in Suz12 *de novo* binding after 72 h depletion of Ring1B levels and the subsequent 24 h induction of Ezh2, indicating that H3K27me3 and H2AK119ub1 are not the only mediators of PRC2 recruitment, as Suz12 and other non-core subunits can bind some of their targets in their complete absence. These data were consistent with prior studies where the global loss of H2AK119ub1 resulted in a strong but not total impairment of Suz12 binding and H3K27me3 deposition [67,76,90,91]; and findings in which the double Ezh1/2 KO did not change global levels of H2AK119ub1 in mESC [20]. All this evidence indicates that Polycomb employs different *de novo* recruitment mechanisms when H3K27me3 and H2AK119ub1 are unavailable.

Additionally, it has been demonstrated that non-core proteins such as the Jumonji and AT-rich interaction domain 2 (Jarid2), the adipocyte enhancer-binding protein 2 (Aebp2), the Polycomb-like protein Pcl2 (Pcl; also known as Mtf2), and those belonging to the Elongin complex (EloB, Elonging BC and Polycomb-associated protein (Epop), and EloC) participate in PRC2 *de novo* recruitment, based on evidence from Native ChIP-mass spectrometry (ChIP-MS) of PRC2 complexes stalled at nucleation sites [31]. Such proteins are crucial elements for the classification of PRC2 subcomplexes. The PRC2.1 complex is associated with Epop and some DNA-binding proteins, such as the Pcl subunits, whereas PRC2.2 can interact with H2AK119ub1-recognizing proteins such as Aebp2 and Jarid2 [69,91–96]. In line with these findings, the depletion of Jarid2 or Mtf2 caused the loss of PRC2 genome-wide binding after 12 h of Eed reinduction. However, Jarid2 alone could not restore the Suz12 binding at nucleation sites after 24 h of Eed induction. Still, the binding of PRC2 was recovered by the eighth day if Jarid2 or Mtf2 were present to stabilize the complex at chromatin [31]. Previous work has shown that Mtf2, Jarid2, and Pcl3, but not Epop, were recruited to their targets in Ezh2-depleted cells [20]. These pieces of evidence imply a preponderant role of Mtf2 and Jarid2 in *de novo* recruitment mechanisms of PRC2 subcomplexes (Figure 2).

Exciting work has shown that PRC2.1 and PRC2.2 have different *de novo* targets during cell differentiation. For example, Auxin Inducible Degron (AID) systems allowed the study of the recruitment of Suz12, Mtf2, and Jarid2 during mESC specification to neural progenitor cells (NPC), where Suz12 elimination causes severe reductions in H3K27 methylation at 24 h treatment and impairs NPC specification [97]. In the absence of Suz12, the neural genes *Fabp7*, *Nestin*, and *Sox1* were not overexpressed, nor were pluripotency markers inhibited, corroborating that Suz12 is necessary for establishing new Polycomb patterns during cell differentiation. On the other hand, Mtf2 and Jarid2 depletion at 2 h treatment does not affect total H3K27me3 levels [97]. The last demonstrates previous observations that individual non-core subunits are dispensable for target site-specific methylation but jointly necessary for Polycomb binding in homeostasis, as each PRC2 auxiliary subunit has some chromatin targeting activity [98].

However, the analysis of the inducible AID-degradation of Mtf2 and Jarid2 before NPC differentiation shows that PRC2.1 and PRC2.2 repress different targets during cell-fate transitions [97]. Thus, PRC2 complexes can act redundantly in the pluripotent state but elicit different repression mechanisms for acquiring a new cell identity. The Mtf2-sensitive genes had a higher CpG-rich density at their promoters and greater H3K27me3 and Suz12 levels at their TSS than the Jarid2-sensitive genes across differentiation. Instead, Jarid2-sensitive genes showed higher H3K4me3 levels in mESC, but the mark decreased after NPC differentiation. Hence, the Mtf2 targets seem to be constitutively silenced genes, while Jarid2-sensitive become repressed during differentiation. Single-cell transcriptomic analyses of embryoid bodies (EB) derived from Mtf2 and Jarid2 KO mESC showed that Mtf2 suppressed poised lineage-specific transcription factors and signaling molecules. At the same time, Mtf2-KO EBs differentiated faster to the three germ layers than the Jarid2-KO EBs [17]. All the above strengthen the idea that Mtf2 ensures the deposition of H3K27me3 at Polycomb key genes, whereas Jarid2 is essential for dispersing new repressive domains.

Recent evidence has indicated that the interaction with Jarid2 and Aebp2 could be disturbed in the complex assembled by the short Suz12 isoform that skips a coding exon on the C2 domain [99]. Similarly, the lack of the ZnB domain on an oncogenic-fused Jazf1-Suz12 form impairs the interaction with Jarid2, Epop, and Pali1, causing ectopic recruitment during cell differentiation [100]. Thus, more research is needed to understand the effects of Suz12 isoform and fused proteins in PRC2 recruitment in homeostasis and disease.

### Mtf2 Is a Key *De Novo* Recruitment Partner in mESC

Different proteins have been studied during *de novo* recruitment (Table 1). Among all Polycomb-like proteins, Mtf2 is the one that has attracted the most attention because of its capability to bind CG-rich DNA and persist in chromatin regardless of astringent conditions [101]. Mtf2 recruits PRC2 independently of other non-core subunits and are less affected by the H2AK119ub1 depletion or KO for Ezh2 and Jarid2 [91,98]. Even so, Eed inhibition reduces 85% of the Mtf2 recruitment in mESC [102], indicating that a minor proportion of the Mtf2-binding is independent of the write-and-read mechanism. Conversely, the Mtf2-KO severely affects the Polycomb regulatory feedback loop as Mtf2 KO reduces Suz12 and H3K27me3 levels comparable to Ring1B KO [69]. In addition, the double KO of Mtf2/Ring1B completely displaced Suz12 from its target promoters in mESC [69], meaning that PRC2 recruitment relies on Mtf2 when the positive feedback loop is impaired. Genome-wide, more than 70% of Ezh2-binding sites and H3K27me3 peaks show reduced levels in Mtf2-depleted cells, which are more significant than in the Jarid2 KO condition [17,50,102]. The latter suggests that the PRC2.1 complex leads to PRC2 nucleation site formation [17,102].

**Table 1.** The participation of Polycomb subunits in *de novo* recruitment and domain formation.

| Complex | Subunit | Key Role | Predominant Expression | Ref |
|---|---|---|---|---|
| cPRC1 | Pcgf4 | Participates in LLPS and *de novo* condensate formation in artificial inducible systems. | Differentiated | [19,67,76,103,104] |
| | Cbx2 | Participates in LLPS physiologically | Differentiated | [56,105,106] |
| vPRC1 | Phc1 | Participates in LLPS and *de novo* condensate formation in artificial inducible systems. | Pluripotent | [107–110] |
| | Pcgf1 | Bridge for Kdm2b-mediated recruitment and nucleation site formation. | Pluripotent | [27,67,76,111–114,114–117] |
| | Pcgf3 | *De novo* recruitment through the interaction with hnRNPK and ncRNAs. | Pluripotent | [75–77,118–124] |
| | Pcgf5 | *De novo* recruitment through the interaction with hnRNPK and ncRNAs. | Differentiated | [76,77,118–124] |
| | Pcgf6 | Implicated in *de novo* recruitment and domain nucleation. Interaction with DNA-binding factors (e.g., Max/Mga). | Pluripotent | [9,61,69–73,76,79,81,113,125,126] |
| | Rybp | Necessary for cell proliferation, H3K27me3 maintenance, and H2AK1191ub spreading. | Pluripotent | [22,61,66,81,127,128] |
| PRC2.1 | Kdm2b | Binding to non-methylated DNA. The long isoform Kdm2b mediates *de novo* recruitment at the peri-implantation stage. | Pluripotent | [4,11,30,112,129] |
| | Epop | Serves as a bridge for the interaction with the ELOBC. | Pluripotent | [69,85,130–132] |
| | Pcl1 | Contributes to PRC2 recruitment at narrow Polycomb domains. | Differentiated | [50,91,93,101,133–138] |
| | Mtf2 | *De novo* recruitment thought CGI binding at nucleation sites. | Pluripotent | [50,101,133,139,140] |
| | Pcl3 | Contributes to PRC2 recruitment at narrow Polycomb domains. | Differentiated | [13,85,101,133,137] |
| PRC2.2 | Aebp2 | Stimulates PRC2 catalysis and recruitment to methylated DNA in vitro. Promotes PRC2 occupancy on chromatin. | Equally expressed | [96,141,142] |
| | Jarid2 | Recruits the complex to chromatin by recognition of H2AK119ub, or the CGIs. | Pluripotent | [4,86,94,143–147] |

Abbreviations. ELOBC: Elongin BC complex; CGI: CpG islands; hnRNPK: Heterogeneous Nuclear ribonucleoprotein K; LLPS: liquid–liquid phase separation; ncRNAs: non-coding RNA.

However, not all CpG-enriched regions have the same sensitivity to Mtf2 depletion [97]. The enrichment of GCG at unmethylated CGIs favors a helical shape and recruitment of PRC2 through a stronger association of the Mtf2 EH domain [17,50]. Such motives are copious in narrow repressive domains and coincide with the description of the "GCN" nucleation sites [31,97]. The Mtf2 KO causes an H3K27me3 decrease in 73% of sharp and 44% of broad repressive domains [102]. Moreover, Mtf2 is nearly eliminated at broad regions in Jarid2 and Eed deficient cells, suggesting that the loss of H3K27me3 dispersal restricts the Mtf2 to their DNA-binding targets [50,102]. Although Mtf2 is also present at the broad regions in response to H3K27me3 and the tri-methylation of histone H3 on lysine 36 (H3K36me3) through its Tudor domain, the sharp domains, which usually contain bivalent genes, rely more on PRC2.1 *de novo* recruitment [102]. In parallel, Jarid2 attachment may depend on H2AK119ub1 recognition by its ubiquitin interaction motif (UIM) and a more localized PRC2.2 *de novo* recruitment that relies on the CGG and GA sequence motifs [23,94,148]. All these shreds of evidence support the participation of Mtf2 in PRC2 *de novo* recruitment in mESC (Figure 2). Nevertheless, our knowledge about how the tissue-specific stoichiometry of subunits affects overall Polycomb function and *de novo* targeting is still limited. Elucidating the mechanisms regulating *de novo* recruitment is essential to understanding the nature of Polycomb-mediated rewiring during cell differentiation.

### 3.3. RNA Can Act as a Bridging Element between CGIs and Polycomb

A clear link between RNA and Polycomb targeting has been suggested in pioneering studies regarding X chromosome inactivation in mice [149,150]. However, a direct RNA-mediated recruitment is still controversial due to the variety of alternative regulation mechanisms in which Polycomb binds to nascent or free RNA molecules that facilitate or prevent their engagement and catalytic activity in chromatin [119,151–153]. There is evidence of plentiful Polycomb–RNA interactions that are "weak in specificity but strong in affinity" [154] plus dual regulatory functions suggesting that RNAs can serve as bridging elements or fine-tune regulators of Polycomb activities. For example, RNA immunoprecipitation (RIP-seq) experiments have shown that thousands of transcripts are associated with core and non-core subunits such as Ezh2, Suz12, Aebp2, and Jarid2 and that some DNA/RNA duplexes, called R-loops, can stabilize Ezh2 and Ring1B binding to chromatin [155–158]. On the other hand, GC-rich sequences such as the G-quadruplex structures (G4s) present at nascent RNA induce PRC2 eviction from chromatin [159–161]. The interaction of Polycomb with G4s or chromatin is mutually exclusive and restricts PRC2 residency depending on the transcriptional status of each target [161]. Thus, a high transcriptional status could be associated with G4s interaction, the inhibition of PRC2 methyltransferase activity, and the impairment of substrate recognition through sterical obstruction of its catalytic lobe [146,160,161]. Meanwhile, low transcriptional status may allow PRC2 binding to chromatin [25,160–162].

The above does not rule out the existence of RNA-specific mechanisms to recruit Polycomb. One of the best-characterized examples of the role of an RNA molecule in PRC2 recruitment is the long non-coding HOX transcript antisense RNA (HOTAIR) that physically interacts with PRC2 and participates in the repression of the *HOXD* locus, being one of the best archetypes for RNA-mediated recruitment of Polycomb [163–165]. However, there is controverting evidence that PRC2 recruitment is a response to gene silencing and that PRC2 is dispensable for the repression at the *HOXD* locus because the ablation of the Eed or Suz12 causes no changes in transcriptional repression compared to the WT [25,154]. Thus, apart from physical interaction and coexistence in each locus, time-lapse studies could help reinforce hypotheses about RNA-mediated recruitment.

However, the quintessential example of RNA-mediated recruitment of PRC2 is the long non-coding transcript Xist [150,166], which coordinates with accessory proteins to initiate Polycomb domain formation during monoallelic chromosome silencing in mammalian females [167]. The interaction between Xist and Polycomb relies on the Xist RNA Polycomb Interaction Domain (XR-PID) circumscribed in the B and C-repeat elements of Xist as their alteration impairs *de novo* recruitment [118–121,124]. Mechanistically, XR-PID associates with the Heterogeneous Nuclear ribonucleoprotein K (hnRNPK), which has a leading role in Pcgf3/5 recruitment and the subsequent spreading along the chromosome that supports the topological configuration of the X chromosome [118,120,146,156,159,160,166–168]. Thus, hnRNPK acts as a bridge between RNA and vPRC1, corroborating the participation of non-coding RNA (ncRNAs) and accessory proteins in Polycomb *de novo* recruitment (Figure 2). Remarkably, H2AK119ub1 is critical in the RNA–Polycomb silencing as Ring1A/B-depleted cells impair Xist-mediated PRC2 recruitment. Furthermore, pharmacological interference of H2AK119ub1 deposition caused a reduction in Eed binding and H3K27me3 deposit, indicating that H2AK119ub1 precedes Xist-mediated PRC2 recruitment [167]. Interestingly, from all Pcgf subunits, only the Pcgf3/5 were retained in Xist domains after photobleaching, indicating a stable interaction of vPRC1 and this RNA molecule. Concomitantly, just Pcgf3/5 double KO decreases the levels of the repressive marks at Xist domains, leading to the conclusion that H2AK119ub1 deposition by vPRC1 initiates the Xist-dependent recruitment of Polycomb [167]. Moreover, the kinetics of histone deacetylation, gene silencing, and H2AK119ub1 deposition are intimately correlated. Histone deacetylation is essential for the transcriptional silencing and initiation of Polycomb binding to intergenic regions [169]. At these sites, the rapid deposition of H2AK119ub1 antecedes Xist coating and H3K27me3 dispersion.

Other lncRNAs have been implicated in Polycomb recruitment, such as Anril, Braveheart, and Fendrr [170–173], and in particular, ncRNAs such as Meg3, Airn, and Kcnq1ot1 also interact with hnRNPK and may be able to form Polycomb nucleation sites at CGIs in autosomes [122,123,145,174]. Hence, hnRNPK may mediate Polycomb recruitment via RNA in more than one gene (Figure 2). The removal of hnRNPK caused reductions in H3K27me3 and cell death after two days of ablation in mouse trophoblast stem cells (TSCs) [122]. Interestingly, the number of hnRNPK–RNA complexes ranges between cell types, although the comparable levels of PRC2 subunits and availability of the ncRNAs [122,123]; thus, there may be different mechanisms regulating the lineage-specific stoichiometry of the RNA–Polycomb complexes, but more studies are necessary to understand their cooperation in *de novo* recruitment.

## 4. *De Novo* Domain Dispersal Depends on the Polycomb Allosteric Activation and Closeness between Nucleation Sites

Diverse evidence has led to the proposal of a "nucleation and dispersion model" for *de novo* PRC2, which states that initially, PRC2 recognizes its nucleation sites and catalyzes H3K27me2 [31,56,175–178]. Once the nucleation site has reached a significant threshold of PRC2, lysine 27 is trimethylated. From there, PRC2 binds to its catalysis product, H3K27me3, and undergoes an allosteric activation mediated by the Eed cage domain [31,45,179,180]. Finally, the H3K27me2 mark is converted to H3K27me3 proximally and distally through long-range contacts provided by its counterpart, PRC1. As PRC2 moves away from its nucleation sites, its presence in chromatin decreases and only deposits H3K27me2, concomitantly with the evidence that only H3K27me2 spans large domains across the genome [31,181,182].

Furthermore, experiments based on the catalytic inhibition of Ezh2 have confirmed that most of the early deposited H3K27me1 lie in regions that become di- and tri-methylated. Still, its habitual targets reach steady-state levels later [32]. Thus, histone modification occupancy changes after Polycomb reintroduction in virtue of time.

Congruently with the nucleation and dispersion model, the Eed-carrying mutation at the aromatic cage domain becomes stuck at the nucleation sites and fails to reconstruct the spreading areas characterized by lower CpG counts and less enrichment of Suz12 binding [31]. Elegant in vitro oligonucleosome array assays have shown that mutant cage Eed cannot propagate H3K27me3 in cis from the pre-modified nucleosome [31]. However, when the Eed endogenous protein is reintroduced, the time window for domain rebuilding is shortened. The reintroduction of Eed for 12 h showed H3K27me3 recovery at the same sites retained by the Eed cage mutant. After 36 h, the methylation distribution reached the steady-state condition [31]. All the above demonstrate that the *de novo* recruitment sites are nucleation hot spots that depend on the Eed aromatic cage and PRC2 allosteric activation to disperse the repressive domains.

Ezh2 auto-methylation is another necessary factor to attain *de novo* H3K27me3, as demonstrated by the inability to resemble the H3K27me3 patterns when the auto-methylation loop of Ezh2 is mutated [183]. Thus, automethylation is essential for the complete allosteric activation of the enzymatic activity of Ezh2. Besides, it is unlikely that Ezh1 alone, as to the Eed cage mutant protein, can spread *de novo* H3K27me3 across the boundaries of PRC2 nucleation sites as Ezh1 sustains the regular deposition of H3K27me1 but reduced H3K27me3 levels in the absence of Ezh2 in a human cell line [20,31]. The failure of Ezh1 to spread the mark may result from a lower allosteric stimulation than Ezh2 [142,184]. This behavior could explain why Ezh2 is more relevant in proliferating cells, where the active write-and-read spreading mechanism is needed. In differentiated post-mitotic cells, the crosstalk between repressive complexes and Ezh1 is sufficient to maintain the methylation patterning [185].

Other factors, such as allosteric competitors, impair Polycomb dispersion kinetics in disease. For example, PRC2 can be interfered with by histone H3K27M mutation or the inhibitor Ezhip [186,187]. These oncoproteins act as PRC2 competitive inhibitors,

reducing H3K27me3 deposition and spreading from CGIs [188]. Despite losing H3K27me3 in intergenic regions, the CGIs retain PRC2 affinity and activity [188]. Inducible models have shown that Ezhip stabilizes the complex at CGIs, obstructing the spreading but allowing the formation of H3K27me3 narrow peaks. This evidence suggests that the sites where H3K27M and Ezhip stall the PRC2 complex may coincide with the *de novo* recruitment sites, although this is an unproven hypothesis. Hence, an example of how Polycomb dysregulation, particularly *de novo* recruitment, renders the development of certain disorders such as cancer [189–193]. Similarly, the Eed mutant protein found in the Weaver syndrome is deficient in spreading H3K27me3 [142], potentially explaining the genetic deregulation in the disease. Thus, understanding Polycomb recruitment and domain dispersal is a long-standing interest in the field.

## 5. Polycomb Nucleation Sites Contact High-Ordered Structures

The fact that Polycomb spreads from a discrete number of zones highlights the importance of each nucleation site. Indeed, discrete nucleation sites can affect in cis the distal dispersion of Polycomb; this was demonstrated by eliminating the *Evx2* promoter from which H3K27me3 spreads across the *HoxD* cluster [31]. After 24 h of Eed reinduction, a lower signal of H3K27me3 and delayed deposition was detected across the *HoxD* cluster, although levels recovered eventually. The delay is strongly influenced in cis, as the H3K27me3 recovery takes less time if another nucleation site is near the deleted site. Thus, if the proximity from a pre-modified nucleosome can influence the spreading of H3K27me3 to the neighborhood, it seems possible that methylation could also extend to distant genomic sites through the long-range interactions of Polycomb domains.

Some nucleation sites are physically in contact with each other. Through 4C sequencing (4C-seq) analyses, researchers observed that the *Evx2* targeted site interacts with 11 Polycomb nucleation sites in the same chromosome, advising that Polycomb clustering may facilitate H3K27me3 spreading [31]. The interactome of the *Evx2* promoter is independent of the catalytic activity of PRC2, as mutant cage Eed only reduces the interaction frequency of two nucleation sites. So, additional elements may be regulating the strength of these interactions, as the oligomerization of some cPRC1 subunits, such as Pcgf4 and Phc1, can re-amplify the Polycomb preexisting foci or cause *de novo* condensates [104,105,194].

The fact that nucleation sites spatially contact each other is relevant if we consider the existence of the Polycomb bodies, perichromatin foci characterized by their 3-dimensional arrangements decorated with Polycomb and H3K27me3 [195]. Intriguingly, immunofluorescence in situ hybridization experiments in mESC showed an overlap between the *Evx2* nucleation site and the earliest foci of H3K27me3 that appeared shortly after the reintroduction of Eed [31]. However, these early H3K27me3 hubs do not appear in the Jarid2/Mtf2 double KO, corroborating their role in PRC2 *de novo* recruitment. Then, Polycomb bodies may coincide with the nucleation sites of repressive domains (Figure 3). Intriguingly, many nucleation sites found in Eed rescue experiments overlap with the methylation peaks that appear in the inner cell mass and increase in number during the transition to the epiblast stage [31,196]. Polycomb bodies have different locations inside the nucleus depending on the physiological context [2,197]. In sum, the number and position of Polycomb bodies varies on the cellular lineage and developmental stage as to the location of nucleation sites, in support of the Polycomb cell-specific *de novo* recruitment model.

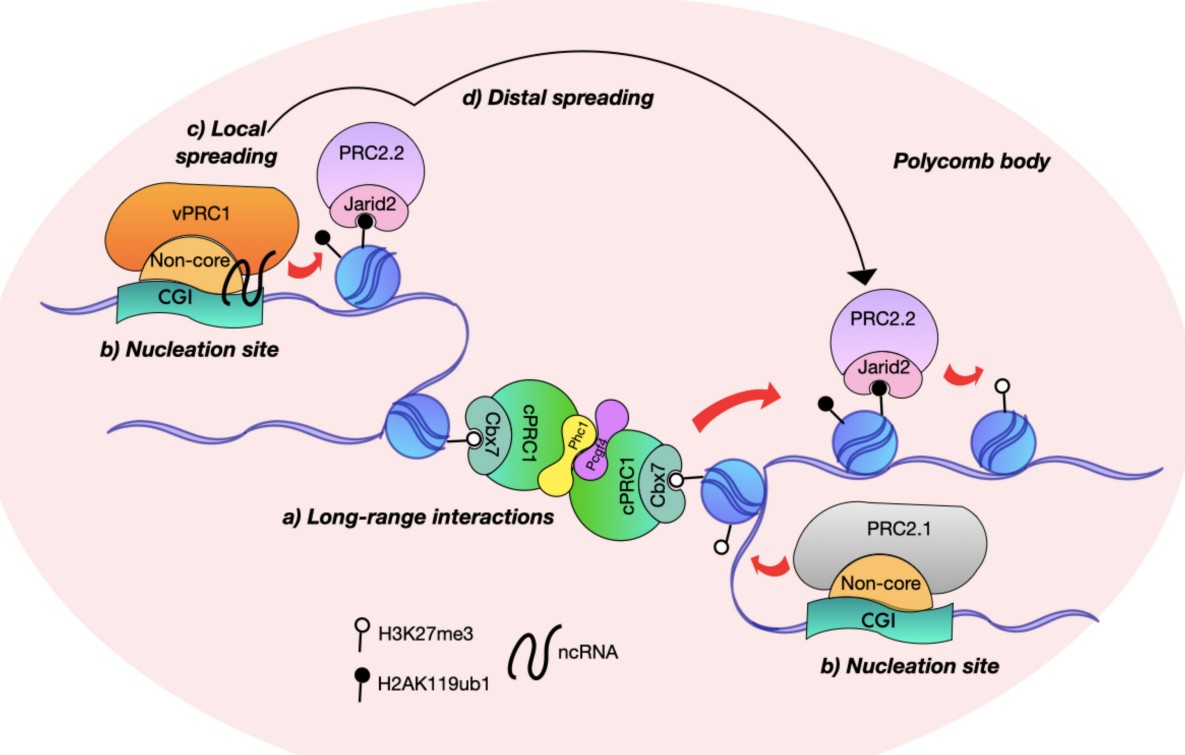

**Figure 3.** The nucleation sites interact at the Polycomb bodies. (**a**) The cPRC1 complexes sustain Polycomb long-range interactions through the oligomerization of Phc, Cbx, and Pcgf subunits. Please refer to Table 1 for more information about the role of each subunit; (**b**) the Polycomb contacts coincide with the nucleation sites, from which the spreading of repressive marks takes place; (**c**) local spreading acts in cis, from one modified nucleosome to the next; (**d**) the vicinity favors the distal spreading with distant nucleation sites because of the long-range interactions and Polycomb concentration.

*The Interplay of Architectural Proteins at the Polycomb Contact Sites*

Polycomb bodies are contact sites for nucleation and methylation dispersal that also participate in nuclear topology organization (Figure 3). Polycomb loop anchor points coincide with the nucleation sites of Eed mutants [31], and a third of these loops are also present in human-induced pluripotent stem cells (iPSC), implying that some Polycomb interacting sites are shared among species and cell types [198]. In mESC, hundreds of H3K27me3-associated DNA loops are present in all chromosomes bringing regions of tens of megabases long together [198].

The architectural role of Polycomb contacts is affected by the catalytic activity of PRC1 as constitutive or inducible systems of non-catalytic Ring1A/B subunits have exposed the consequences of losing H2AK119ub1. For example, capture-C experiments showed that the 72 h induction of catalytic-dead Ring1B on a Ring1A KO background impaired the interaction profiles of 24 Polycomb target genes to a similar extent as the complete removal of PRC1 [72]. Furthermore, PRC1 catalytic impairment significantly reduced the number, size, and fluorescent intensity of Polycomb bodies [72]. This evidence indicates that H2AK119ub1 deposition catalyzed by vPRC1 is required to recruit cPRC1, which handles long-range interactions that promote repressive domain dispersal and chromatin compaction.

The importance of Polycomb-associated interactions is undeniable, as the loss of long-range interactions alters cell-specific genome organization and the expression of pluripotency genes in mESC [108,199]. Optical Reconstruction of Chromatin Architecture

(ORCA) assays showed that deletion of the anchor sites affected the physical organization and H3K27me3-spreading of the regions adjacent to the loops, similar to the removal of Polycomb nucleation sites [31,198]. However, it is unclear how genome-folding and chromatin reorganization during differentiation relates to Polycomb binding.

In line with the nucleation and dispersion models, PRC1-enabling long-range interactions could participate in a new *de novo* recruitment mechanism, in which the initial binding of vPRC1 guides the formation of cell-specific chromatin contacts where PRC2 is engaged to turn them into nucleation sites. Together, DNA and chromatin contacts provided by PRC1 could be the constant *de novo* signal that allows Suz12 to reconstruct Polycomb domains when repressive histone marks are not present. However, several molecular players are involved in the 3-dimentional organization of the genome, and future studies should reveal to what extent PRC1 long-range interactions change the accessibility and formation of the PRC2 nucleation sites for a given cell lineage.

Recently, the connection between some architectural proteins and the maintenance of Polycomb contacts has been explored. Initially, researchers suggested a domain barrier function of Ctcf due to its binding at the boundaries of repressive Polycomb domains [200,201]. Still, it is thought that the disruption of Polycomb domains is independent of Ctcf insulation. Capture-C experiments showed that Ctcf-AID removal did not affect distal Polycomb interactions [202]. Therefore, Ctcf contributes little to Polycomb interaction disruption in mESCs. Furthermore, the association between Ring1B and its CGI targets is unperturbed upon Ctcf degradation [202]. Other assays with Ring1A KO cells showed that TADs were unaffected by auxin degradation of Ring1B. In contrast, Polycomb interactions were lost in Ring1A/B-depleted cells, corroborating the role of PRC1 in maintaining Polycomb long-range interactions [203]. Thus, the mechanism for the PRC1 long-range interaction is different from that of Ctcf-mediated loop extrusion.

AID systems for cohesin subunit Rad21 revealed that the 6 h treatment with auxin caused a massive loss of TAD in mESC, but the persistence of Polycomb bodies and ~336 interactions across the genome enriched by Ring1B, Ezh2, Suz12, and the histone marks H2AK119ub1 and H3K27me3 [203]. So, the Polycomb cell-specific contacts and nucleation spots seem unaffected by the absence of cohesin. However, long-range interactions between Polycomb domains increase their strength in the absence of cohesin, suggesting that cohesin counteracts PRC1 distal contacts [203]. Moreover, degradation tag systems (dTAG) performed during the middle and late points of chromosome X silencing showed that about a tenth of the constant loops strengthened after 8 h Rad21 degradation, whereas 72 h degradation of the cohesion release factor Wapl increased the cohesin association to chromatin and weakened Polycomb loops spanning more than 1 Mb [204]. So, cohesin retention may affect the interaction strength of Polycomb domains and H3K27me3 spreading, causing the upregulation of the target genes inside the loops.

Similarly, the AID-mediated degradation of the cohesin regulator Pds5A impairs cohesin unloading, increases insulation, and lessens ultra-long chromatin loops among Polycomb targets [74]. Pds5A degradation reduces the interaction frequency of 65 anchor sites that become upregulated. So, the elimination of Pds5A potentially affects the condensation of some Polycomb bodies [74]. All the above suggest that PRC1 availability influences Polycomb high-ordered interactions and that the equilibrium on the eviction of some architectural proteins must affect the contacts and maintenance of nucleation sites. Similarly, the disturbance of other transcriptional regulatory elements, such as the CDK–Mediator complex, decreases the interaction of Polycomb-associated promoters without affecting H3K27me3; however, the process assisting cPRC1 chromatin association is still unknown [194].

Intriguingly, 4C-seq analyses showed that introducing RNA-binding deficient Ezh2 mutants caused a loss of Polycomb-associated interactions [198]. This evidence could have implications for our understanding of the role of RNA in Polycomb *de novo* recruitment and domain formation. However, as these mutants affect the global deposition of H3K27me3, it may indirectly decrease cPRC1 recruitment. Future studies must discern if the loss

of contacts is caused by the interference of RNA-binding functions of PRC2 or by the disequilibrium in the Polycomb feedback loop. At this point, we can contemplate that Polycomb recruitment is essential for the 3D nuclear architecture. Once the cell identity is established, the chromatin context can also guide the Polycomb re-recruitment upon disruption.

## 6. *De Novo* Targeting in Homeostasis and Replication: Interconnection between Complexes

The cell- and tissue-specific patterning of Polycomb domains and the mechanisms behind the recruitment of the repressive complexes have intrigued researchers for years. To date, the evidence suggests that PRC1/2 functions are intertwined in a context-dependent feedback loop as the complexes can be recruited independently. Still, the perturbation of one affects the activity and targeting of the other [22,205–207]. Regardless of the signal, the simultaneous performance of different Polycomb recruitment processes is vital to fine-tune the epitranscriptome and buffer deleterious effects in the inactivity of any subunit during replication, homeostasis, and *de novo* domain formation.

The enzymatic and chromatin-binding activities vary among repressive complexes; even variants of the same complex have multiple regulatory functions and different sensitivity to Polycomb domain erosion. For example, vPRC1 is less dependent on H3K27me3 than the cPRC1, while Ring1B occupies half of its targets despite H3K27me3 loss and maintains similar H2AK119ub1 levels compared to the WT [22,102,202]. Moreover, Eed KO mESC showed that reductions in H3K27me3 diminished Ring1B, Cbx7, and Pcgf2 occupancy. Still, the Yy1 binding protein (Rybp) and Pcgf1 association to chromatin are almost unchanged [22,207]; even the Rybp/Eed double KO cells retain Ring1B binding [128]. In addition, Single-particle tracking (SPT) assays revealed that the ablation of Eed or Ezh2 resulted in a reduction in the size of the chromatin-bound fraction of Cbx7 and Cbx8, corroborating that H3K27me3 deposition influences cPRC1 [208]. Henceforth, H3K27me3 is not required for vPRC1 binding to chromatin but greatly influences cPRC1 recruitment, explaining H2AK119ub1 maintenance in the PRC2-defective cells [16].

Consistent with the indirect effect of PRC2 activity on cPRC1 binding, the reduction in H3K27me3 caused by the triple KO of Pcl1/2/3, decreases Cbx7 binding but does not cause changes in Rybp association to chromatin [91]. Moreover, the Pcl1/3 double KO provokes a marked reduction in H3K27me3 at narrow Polycomb domains, but not over the broad where Jarid2 and Aebp2 remain attached [91]. Thus, the interpretation is that PRC2.2 and cPRC1 contribute to maintaining broad Polycomb domains through the write-and-read mechanism. In contrast, PRC2.1 and vPRC1 have additional engagement mechanisms, such as the *de novo* recruitment at the target promoters that is relevant during cell differentiation.

These recruitment mechanisms affect repressive domain inheritance in homeostasis, where the nucleosomes are disassembled immediately before the replication fork, and the parental nucleosomes are locally re-deposited on the new strands [209–214]. The parental histone recycling allows the Polycomb machinery to restore methylation patterns after replication through the Cbx7 recognition of H3K27me3 [46,179,180].

Elegant tethering systems in mESC have shown that Cbx7-TetR and Rybp-TetR recruitment can induce gene repression [178]. Interestingly, gene repression and TetR-Cbx7 signal persist in the absence of the initial recruitment stimulus, even a dozen division cycles after the release of the chimeric proteins [178,215]. On the contrary, TetR- Rybp release causes gene reactivation and the displacement of PRC2 from chromatin [178]. The latter denotes that cPRC1 has a sequence-independent propagation mechanism, guided by H3K27me3 recognition, that allows it to move across the target site; meanwhile, vPRC1 stabilization at chromatin is necessary to maintain gene repression through H2AK119ub1 deposition. According to this evidence, the H3K27me3 depletion affects repression inheritance but not repression initiation.

In congruence with the H3K27me3-dependence of cPRC1, loss-of-function systems showed that more than half of Rybp-depleted cells retained reporter gene repression after

Cbx release. Still, a minor proportion of Suz12-depleted cells preserve gene repression after Cbx release [178]. Hence, the data indicate that just cPRC1-initiated domains along with PRC2 maintain gene repression after cell division through the write-and-read mechanism.

Nonetheless, other simultaneous mechanisms such as the *de novo* Polycomb recruitment might guarantee the histone methylation patterning after the 2-fold replicational dilution of parental nucleosomes [128]. Exciting work has also insinuated that there is *de novo* methylation during cell replication. Even with the AID-mediated degradation of the histone chaperone Npmi, which compromises parental histone recycling from repressed loci, some H3K27me3 levels endure at bivalent targets, possibly as a product of the new catalysis and dispersion of the mark [214]. Thus, *de novo* recruitment could be a constant process in cell replication and homeostasis.

In analogy, the H2AK119ub1 has a positive feedback loop for cis propagation. Artificial TetR-fused PRC1 proteins and H1-compacted polynucleosome arrays have shown that H2AK119ub1 spreading is enhanced by chromatin compaction, dependent on Yaf2 and Rybp [128]. The Zinc-finger-mutated form of TetR-Rybp cannot recognize H2AK119ub1 impairing propagation [128]. Captivatingly, the Eed KO does not disturb H2AK119ub1 dispersal, so ubiquitination propagation is not strictly committed to the H3K27me3 regulation axis. In another vein, the release of TetR-Rybp caused just a 30% decrease in the H2AK119ub1 levels after three cell divisions [128]. Thus, as in the case of H3K27me3 inheritance, there is a mechanism to lessen the replicational dilution of H2AK119ub1 that could rely on the binding capability of some accessory subunits.

The abrogation of H2AK119ub1 has its repercussions on Ring1B placement in chromatin. For example, in H2AK119ub1-deficient cells, only the Pcgf6 subunits retain their affinity for some exclusive promoters; neither Pcgf2, Cbx7, nor Rybp reflect the same behavior [69]. The latter argues in favor of Pcgf6-vPRC1 involvement in *de novo* recruitment and indicates that cPRC1 is more susceptible to H2AK119ub1 loss than vPRC1 [69,72]. Again, reductions in H2AK119ub1 may affect cPRC1 targeting due to lower PRC2 recruitment. For instance, the inducible double KO of Ring1A/B triggers a genome-wide loss of H3K27me3, which causes occupancy reduction in over 80% of Suz12 and Ezh2 target sites [19]. PRC2.2 seems to be more H2AK119ub1-dependent than PRC2.1 because experiments with Ring1B-deficient cells showed a complete displacement of Jarid2 and Aebp2 at 48 h; meanwhile, Mtf2 and Epop persisted until 72 h at promoters [69,72,91]. This is consistent with the reported affinity of Jarid2 and Aebp2 for H2AK119ub1 [94–96]. All the above denote the potential role of Mtf2 in *de novo* recruitment. All this evidence supports the hypothesis that different elements bridging the PRC2.1 interaction with DNA are players in *de novo* recruitment.

As expected, the conditional triple KO of Pcl1-3 results in the destabilization of other PRC2.1 members such as Epop and Pali1, but no reduction in the binding of PRC2.2 subunits [91], suggesting that PRC2.1 does not affect PRC2.2 recruitment to chromatin. On the contrary, the elimination of Pcl1-3 proteins is correlated with higher binding levels of Aebp2, suggesting that PRC2.2 can bind some of the PRC2.1 targets when absent. However, Jarid2 KO disrupted Aebp2 binding and moderately displaced MTF2, implying that PRC2.1 could be partially disturbed by PRC2.2 [91]. All the above point out the importance of target redundancy and competition between PRC2 subcomplexes and reflect the complex coexistence of Polycomb recruitment axes.

## 7. *De Novo* Recruitment Reveals the First Steps of Polycomb Gene Repression

The temporal resolution and interdependence of PRC1 and PRC2 functions are the main obstacles to distinguishing between causal and collaborative mechanisms of Polycomb transcriptional repression. The exact pathway driving gene silencing is also unclear because most Polycomb functions explain how Polycomb maintains transcriptionally repressed states rather than initiating them. Former research showed that the loss of Eed and Suz12 in mESC caused gene derepression [16,45,216]. However, the exact mechanism is still being investigated as most datasets have been obtained from long-term stable KO conditions and

possibly from different phenotypes/cell passages [25,217]; thus, they are missing details on the first steps of Polycomb targeting mechanisms.

In line with the fact that PRC2-depleted cells maintain a normal phenotype and self-renewal capacities [16,206], Suz12-dTAG systems have shown that 2 h degradation of Suz12 has discrete effects on gene activation [45,218]. Although PRC2 is not involved in the first steps of gene silencing during differentiation, it is key to maintaining long-term cell identity. Short- and long-term differentiation assays induced by RA showed that Suz12 KO guided minor gene expression changes after 72 h cell differentiation induction but was required to maintain proper silencing of lineage and pluripotency-network genes on the ninth day. Moreover, the H3K27me3-negative genes had no significant derepression compared to those lacking Ring1B and H2AK119ub1 signal, supporting the hypothesis that its counterpart, PRC1, is liable for gene repression [18,219].

Just the simultaneous AID-mediated degradation of Rybp and Eed or Suz12 resembles the consequences of losing PRC1 binding to chromatin, such as the reduction in H2AK119ub1, gene deregulation in more than 2600 genes in 24 h, and whiter and more fragmented cell colonies concerning the WT when stained with alkaline phosphatase, a marker to assess pluripotency [207]. Thus, at least in stem cells, the absence of PRC1, but no PRC2, has a faster influence on transcriptional patterns.

It has been debated whether the presence of the Polycomb complexes or the histone mark per se is the leading force in gene repression. It is known that PRC1-deficient cells cannot be maintained as they undergo spontaneous differentiation and lack self-renewal capacity [18,207]. However, the loss of catalytic activity has almost the same effect on gene repression as the complete PRC1 removal [72,90]. The expression of non-catalytic PRC1 complexes leads to phenotypical changes, robust reductions in cell division, and a similar pattern and magnitude of derepression compared to Ring1A/B KO cells (~3000 genes) [45,218,219]. This suggests that H2AK119ub1 deposition by vPRC1 is critical to maintaining cell identity and viability. Interestingly, the synergy among vPRC1 complexes may regulate transcription through H2AK119ub1 deposition because a series of Pcgf-deficient experiments showed that individual losses did not resemble the effects of Ring1A/B removal [67].

The absence of PRC1 has immediate effects on gene repression. Transcriptional profiling from Ring1A KO-Ring1B-AID cells revealed that the number of upregulated genes doubled after 2, 4, and 8 h of auxin treatment [218]. Interestingly, only after 24 h treatment did the expression changes resemble the effects of the long-term absence of PRC1, emphasizing the relevance of time-resolved analysis in the study of Polycomb systems. Those genes that experienced higher derepression at 2 h used to have the lowest expression in WT cells and higher levels of Ring1B, H2AK119ub1, Suz12, and H3K27me3. On the other hand, less depressed genes at 4 or 8 h used to have a higher initial expression in WT and promoter enrichment of RNAPolII and H3K4me3. Then, PRC1 is key to repressing the expression of already low transcribed genes [25]. Thus, PRC1 counteracts with low-level transcriptional signals instead of strong activated states, as only small increases in the expression were shown upon PRC1 removal [218]. Some hypotheses have arisen in the field, such as the low-level expression of Polycomb genes could be a consequence of the capacity of PRC1 to immobilize RNAPolII, possibly by H2AK119ub1 deposition, based on the evidence that Polycomb promoters are rich in RNAPolII Serine 5 phosphorylation (Ser5-P), which is related to initiated or paused RNAPolII [220–222]. However, ChIP-seq data from Ring1AKO-Ring1B-AID showed that 2 h removal of PRC1 promoted RNAPolII new binding at Polycomb target promoters and accumulation of H3K4me3 [218]. The enrichment of paused RNAPolII Ser5-P was restricted to the TSS following PRC1 removal, suggesting that new RNAPolII binding but not release is responsible for quick gene derepression [218]. In conclusion, PRC1 is not initially related to the low-level transcriptional state of its targets, but removal provokes new RNAPolII binding and low expression increases of Polycomb targeted genes.

Finally, PRC2 recruitment at silenced genes is fast, and a lack of transcription may enable the H3K27me3 dispersal and *de novo* PRC2 binding. The 12 h inhibition of RNAPolII Serine 2 phosphorylation (Ser2-P), which is associated with transcriptional elongation, showed an increase in Suz12 binding sites [25]. In vivo transcription capture sequencing (iTC-seq) corroborated the antagonism between Suz12 binding and the transcription of its target genes after drug treatment [25]. Future research based on time-lapse assays may uncover the equilibrium between the binding of the transcriptional machinery and Polycomb recruitment at low-transcription targets in homeostasis and during *de novo* recruitment.

## 8. Concluding Remarks and Future Perspectives

The classical Polycomb recruitment models recognize two main axes of Polycomb regulation, one based on the H3K27me3 reinforcement of cPRC1 recruitment and the other where vPRC1 boosts PRC2.2 recruitment via H2AK119ub1 deposition. It is evident that the overlap between PRC2/cPRC1 and vPRC1 functions maintains the inheritance and regulatory feedback of Polycomb repressive domains, but additional targeting mechanisms underneath have just been explored. The *de novo* recruitment is perceptible during domain neogenesis, reassembling, or dispersal impairment. It is evident that *de novo* Polycomb domain formation must be vital during cell differentiation and development where newly repressed genes acquire Polycomb histone modifications. As the first location for Polycomb targeting, the *de novo* sites offer information about the timescale of Polycomb participation in gene repression and long-range interactions. The study of domain nucleation sites is vital to understanding recruitment routes where RNA and nucleic acid-binding proteins such as Mtf2, Kdm2b, Jarid2, and hnRNPK bridge the interaction of vPRC1 and PRC2.1 at the CGIs of low-level transcriptional targets. All the evidence opposed the existence of a universal *de novo* signal. Instead, redundant mechanisms operate during cell differentiation. Nuclear topology, local chromatin environment, tissue-specific expression of Polycomb, and accessory subunits mediate the coupling and accessibility for repressive complexes, thus summing a layer of complexity. However, the future challenge is to identify the molecular mechanisms orchestrating the initiation of the distinct *de novo* signals that act at different time scales and cell types to enhance long-term gene repression and ultimately define the Polycomb-epigenome of a cell. These findings are highly relevant to understanding the Polycomb regulatory role in embryo development, cell differentiation, adult tissue homeostasis, and diseases where *de novo* domain formation could be affected.

**Author Contributions:** Conceptualization, I.A.H.-R. and V.J.V.; investigation, I.A.H.-R.; writing—original draft preparation, I.A.H.-R.; writing—review and editing, I.A.H.-R. and V.J.V.; visualization-artwork, I.A.H.-R.; funding acquisition, V.J.V. All authors have read and agreed to the published version of the manuscript.

**Funding:** This work was supported by grants PAPIIT-UNAM (IN203820) and CONACYT Ciencia Básica (0284867) to VJV. IA.H-R is PhD student from the Programa de Maestría y Doctorado en Ciencias Bioquímicas, UNAM and received a fellowship from Consejo Nacionalde Ciencia y Tecnología (CONACyT), CVU 886138.

**Institutional Review Board Statement:** Not applicable.

**Informed Consent Statement:** Not applicable.

**Data Availability Statement:** Not applicable.

**Acknowledgments:** We thank Félix Recillas-Targa (IFC, UNAM), Ma. De La Paz Sanchez-Jimenez (Institute of Ecology, UNAM), Nallely Cano (IFC, UNAM) for critical reading of the manuscript. We thank Joseph Tripp for the final English check.

**Conflicts of Interest:** The authors declare no conflict of interest.

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
