# Peer review of "De Novo Polycomb Recruitment and Repressive Domain Formation"

_2075-4655, 2022_

Round 1

Reviewer 1 Report

In this review Hernández-Romero & Valdez, discuss the mechanisms of PcG complex recruitment to chromatin. PcG proteins are a family of transcriptional and epigenetic repressors that are essential for cell fate specification and development. They are generally divided into two complexes, PRC1 and PRC2, each with enzymatic activity for catalysing repressive histone modifications. The topic of PRC1 and PRC2 complex recruitment to chromatin has been one of active study for the last two decades, with some particularly important mechanistic results coming to light in the last 2-3 years. Therefore, this review is timely and of interest to the chromatin field. PcG proteins are also highly relevant to cancer and neurodevelopmental syndromes as each can be caused by mutations in the PcG machinery, this makes PcG mechanisms relevant to a broad audience.

The authors present a detailed update on how each, of the many, PRC1 and PRC2 sub-complexes can be recruited to chromatin, including through histone modifications, transcription factors, RNA and other factors. There is particular focus on de novo recruitment, in this case referring to recruitment in the absence of other PcG mediated histone modifications.

While the topic of the review is timely, its impact is limited by an abundance of minor grammatical errors. Also, the authors have, in many cases, described some experiments in overly detailed ways. They often provide unnecessary details such as number of targets sites/ percentages/ fold decrease etc, which detract from the readability of the review. A review should summarise the literature and the conclusions of papers/experiments for the reader. This is something that can be improved in certain sections (examples below).

Major Comments

·      The authors provide too many details of specific experiments in papers, as mentioned above. They should instead interpret the data and put the result in the context of other work in a way that would be useful for a non-expert reader. Some examples that could be improved are below.

o   Line 301-304.

o   Line 397-407.

o   Line 247-251.

o   Line 290-322. This section should also mention that Hojfeldt et al, 2019 showed each auxiliary subunit of PRC2 has at least some chromatin targeting activity. This could also be acknowledged in Table 1.

o   Line 641-645. This section should also mention the affinity of JARID2 and AEBP2 for H2AK119ub1 which is critical to the discussed result.

·      Table 1 states that Pcgf4 and Phc1 contribute to LLPS but this has only been shown in an artificial inducible phase separation system (ref 101) not in a physiological setting. Cbx2 on the other hand does exhibit LLPS functions physiologically (Plys et al, 2019, Tatavosian et al, 2019). These changes should be made to the table.

·      The section on RNA recruiting PRC2 has a lot of detail on examples of RNA recruiting PRC2, but it should be updated to provide a balanced argument into this controversial topic. Many studies (Riising et al, 2014, Beltran et al, 2016, Beltran et al, 2019, Zhang et al, 2019) have shown that the RNA-PcG interaction is not specific and actually blocks PcG binding to chromatin or catalytic activity. These alternative mechanisms should also be discussed.

·      JARID2 and MTF2s role in de novo recruitment is mentioned throughout, however the authors appear (see other comment on their definition of de novo) to mean non-histone modification associated recruitment. In this context, they do not adequately discuss that both JARID2 and MTF2 can be recruited through their own histone modification reader activities such as JARID2 reading H2AK119ub1 through its UIM domain and MTF2 (and PCL1/3) reading H3K36me3 through its Tudor domain.

Minor Comments

·      PcG and Polycomb are synonyms, the authors alternate between these throughout and should instead stick with one.

·      The authors focus heavily on de novo PcG recruitment, they should define clearly and specifically what they mean by this. It appears they mean histone modification independent, but this usage seems to vary throughout the manuscript.

·      Line 70-72. This should say ‘de novo recruitment signal can be independent of…’ as there are many cases where recruitment is dependent on catalytic activity.

·      Line 197- the authors could also include Mga, the dimerisation partner of Max, in the list of Pcgf6 recruiting transcription factors.

·      Line 219-220 are over interpretive; it is clear from studies (Hojfeldt et al, 2019, Blackledge et al, 2020, Kalb et al, 2014, Cooper et al, 2016) that there are multiple factors contributing to PRC2 recruitment, including H2AK119ub1.

·      Line 245- This sentence should be rephrased as there is no data published suggesting Epop directly binds DNA.

·      Line 247- the authors should include the references that defined AEBP2 and JARID2 affinity for H2AK119ub1 (Kalb et al, 2014, Cooper et al, 2016, Kasinath et al, 2021).

·      Line 294-295. The cited paper does not contain any Mtf2 ChIP in Eed KO or Eed inhibited cells.

·      Table 1- Aebp2 has been shown to promote PRC2 residency time on chromatin in this paper (Lee et al, 2018), potentially suggesting de novo recruitment, this could added to the table.

·      Line 431- as the discussed Eed mutation is just one of several mutations in the PRC2 core subunits (EZH2, SUZ12 and EED) in Weaver and Weaver-like syndromes this sentence should be rephrased to ‘potentially explaining the genetic deregulation in the disease’, in order to be less definitive.

·      Line 678- the authors describe colonies from a specific experiment as ‘whiter’ without mentioning the stain, or that the stain (alkaline phosphatase) is a marker of pluripotency. This information needs to be mentioned to be informative to the reader.

·      Line 396 ‘Thus, chromatin occupancy of the core and non-core subunits changes after their reintroduction in virtue of time.’ This is not shown in the cited paper. The time course ChIP experiments in that paper only examined histone modifications, not PRC2 subunit occupancy.

·      Line 362-363, there is no pharmacological inhibition of H2AK119ub1 deposition in the referenced paper, only inducible KO systems of PcG proteins.

Typos/ Grammar errors- This is not an exhaustive list of the grammatical errors. The manuscript should be thoroughly checked for other errors to improve readability.

·       Line 10- Disconcerting is not the correct word here.

·       Line 27- ‘tri-methylation’ instead of ‘three-methylation’.

·       Line 32- ‘embryonic’ instead of ‘embryo’.

·       Line 38- ‘Polycomb’ instead of ‘Polycom’.

·       Line 42- ‘are established’ instead of ‘establish’.

·       Line 51/52- sentence not clear, consider re-writing.

·       Line 64- ‘labelled’ instead of ‘label’.

·       Line 68- I am not sure if ‘definition’ is the right term here.

·       Line 77- I think this sentence should finish in ‘PcG mediated histone modifications’?

·       Line 84- ‘editing’ instead of ‘edition’.

·       Line 90 and line 462- ‘3-dimensional’ instead of ‘threedimensional’.

·       Line 113- ‘binding’ instead of ‘union’.

·       Line 140- ‘not’ instead of ‘no’.

·       Line 150- remove the words ‘free signal’ to clarify this sentence.

·       Line 152- ‘enhancer’ instead of ‘enhacer’.

·       Line 179- ‘recruited’ Instead of ‘relucted’.

·       Line 191- ‘directs’ instead of ‘direct’.

·       Line 203- ‘definitively’ is used incorrectly. Consider replacing with, ‘in all likelihood’.

·       Line 234- ‘not’ instead of ‘no’.

·       Line 235- remove ‘in turn’ to improve sentence clarity.

·       Line 252- ‘not’ instead of ‘no’.

·       Line 293-294- should be rephrased that ‘Mtf2 recruits PRC2 independently of other non-core subunits.’ A feature which the cited paper shows is not unique to Mtf2.

·       Line 309- I am unsure what the authors mean by ‘parallel with proof’, consider removing for clarity.

·       Line 386- ‘lysine’ instead of ‘lysin’.

·       Line 397 – ‘carrying’ instead of ‘caring’.

·       Line 398-I am unsure what the authors mean by ‘stock’- perhaps this should be ‘become stuck’ or ‘stop’?

·       Line 441- ‘recovery’ instead of ‘recovering’.

·       Line 441- the word ‘site’ is missing from end of sentence.

·       Line 442- changing ‘closeness’ to ‘proximity’ would improve the clarity of the sentence.

·       Line 471- The word ‘knowingly’ doesn’t make sense in this context.

·       Line 472- ‘nucleus’ instead of ‘nucellus’.

·       Sentence 480-482 seems incomplete? Or it doesn’t make sense in its current format, consider revising.

·       Line 510 the phrase ‘has blown in’ does not make sense in this context.

·       Line 516- ‘targets’ instead of ‘target’.

·       Sentence on line 523-524 does not make sense. What does it mean that PADs are related to 45% of the genome? This sentence needs clarification, are the authors saying that 45% of the genome is not in A or B compartments?

·       Line 533- ‘absence’ instead of ‘lack’.

·       Line 540- ‘the’ instead of ‘de’.

·       Line 565- ‘Amazed’ does not seem like the appropriate word. Maybe the authors could use ‘intrigued’ instead?

·       Line 566- ‘context-dependent’ instead of ‘context-depend’.

·       Line 629- ‘axis’ instead of ‘ax’.

·       Line 690- ‘deposition’ instead of ‘deposit’.

·       Line 712- ‘not’ instead of ‘no’.

·       Line 718- I think this is a mis-reference. The paper listed as #224 does not include any of the mentioned experiments, and is focussed on CTCF not PcG.

·       Line 252- Epop has already been mentioned on line 246, the full name should be defined at its first appearance in the text.

·       Line 634-635 mentions Pcgf6 twice, this looks like a typo?

Reviewer 2 Report

The review by Hernández-Romero I. A. and Valdes V. J. discuss the repressive roles of Polycomb proteins in gene regulation and especially focusing on how the repressive complex proteins find their targets. In this manuscript the authors discuss the earliest steps of Polycomb domain formation and the step wise temporal details of the following transcriptional regulation. They have done a comprehensive study on the topic. I have following suggestions that I believe will help improving the manuscript:

Comments:

1.       The figures are not self-explanatory and figure legends are not explained very well.  For example:

a.       In figure 1: the CGI, cPRC1 and vPRC1 are not expanded/explained in the figure legends.

b.       Similarly Cbx7 and Jarid2 are also not mentioned in Figure legends. If the authors don’t prefer to highlight the proteins here, please refer to the table in figure legends so one could understand what the role of these proteins is.

c.       Histone marks with empty and full circles are mentioned in legends but what are the half-filled/half-empty histone marks?

d.       The existence of H3K27me3 and H2AK119ub1 should also be shown on same histone complex. From the models in all three figures, it appears that different PRC complex proteins are recruited to different histones and never on the same histone.

e.       In figure 2: What is the Y-axis i.e. peak height representing? What is the Gen X label and the small rectangle on the Y-axis or is it a gene?

2.       The acronyms should be explained on their first appearance, for example, CGIs are expanded on page 3 while it appears as early as in abstract. Similarly vPRC1, cPRC1 etc.

3.       In abstract, “ Every cell of an organism shares the same genome; even so, each cellular lineage owns a 8 different transcriptome”, could be “ Every cell of an organism shares the same genome; even so, each cellular lineage owns a 8 different transcriptome and proteome”.

4.       On page 1, line 38, Polycom should be Polycomb.

Reviewer 3 Report

The authors present a thorough review of Polycomb-group recruitment in mammals that will be a useful resource to others in the field. I have two general comments that I hope will further improve its usefulness. 

Specific Comments:

1. The authors should carefully re-check the papers they are citing. For example, ref. 35 (line 98) describes the role of Combgap in PcG recruitment. Other papers describe the role of Pho. Also, although their binding sites are present in many PREs, neither Zeste nor GAGA factor, unlike Pho, has been shown to recruit PcG proteins (line 97). Another example is the citation of Wang et al. (ref. 224) for the effect of inhibition of RNA polymerase II phosphorylation on PcG recruitment. A more appropriate citation for this observation is Riising et al. (ref. 25). These are two examples, but I cannot rule out that other citation corrections are needed. 

2. X-inactivation arguably is the best studied example of de novo PcG recruitment in mammals. In particular, Zylicz et al. (2019) Cell 176: 182-197 provides a detailed description of the temporal order of histone modifications and PRC1/2 during X-inactivation. I would suggest adding a section on this topic.  

Round 2

Reviewer 1 Report

The authors have adequately addressed all comments and improved the review accordingly.